# "To Live Is a Matter of Time": Memory, Survival and Queer Refugeehood in Ocean Vuong's *On Earth We're Briefly Gorgeous*

Sara Soler i Arjona 

Department of Modern Languages and Literatures and English Studies, Faculty of Philology and Communication, University of Barcelona, 08007 Barcelona, Spain; sara.soler@ub.edu

**Abstract:** In his novel *On Earth We're Briefly Gorgeous* (2019), Ocean Vuong attempts to reweave the historical threads that have been brutally severed by American imperialism, forced migration and the imperatives of assimilation, as a practice of survival. Drawing on his own experience as a Vietnamese refugee, Vuong situates a Vietnamese American queer protagonist at the centre of his non-linear narrative, which excavates the boy's family history to trace the multiple histories of displacement informing who he is today. The novel's temporal disorientation becomes a formulation of queer temporality that activates a critical reorientation of how experiences of refuge are typically represented—a coming into consciousness known as "refugeetude". Such a critical reorientation serves a dual purpose. Firstly, by foregrounding the protagonist's—and his family's—shattered recollections, Vuong challenges dominant accounts of the Vietnam War and recovers the voices of those that are effaced by Western representation, thus assembling a more inclusive "just memory" of the war. Secondly, the novel disrupts the teleological narrative of progressive assimilation that is prevalent in refugee discourse by revealing the enduring forms of violence that displaced subjects must still face in contemporary America. By queering the normative temporality of refugee experience, the novel demonstrates how the characters' refugeehood is not finite but ongoing, necessitating a continuous search for healing and resilience.

**Keywords:** queer refugeehood; just memory; queer temporality; refugeetude; queer literature; refugeehood; refugee narratives; survival

## 1. Introduction

Vietnamese American writer Ocean Vuong is known for his 2016 lauded poetry collection *Night Sky with Exit Wounds*, which won both the T.S. Eliot Prize for Poetry and the Whiting Award. Published in 2019, his debut novel *On Earth We're Briefly Gorgeous* not only achieved New York Times bestseller status but also received widespread critical acclaim, emerging as a finalist for the 2020 PEN/Faulkner Award for Fiction and being longlisted for the 2019 National Book Award for Fiction. Vuong's ascent aligns with a growing interest in writers from the Vietnamese diaspora, exemplified by Viet Thanh Nguyen's Pulitzer Prize-winning novel *The Sympathizer* (2015) and Diana Khoi Nguyen's National Book Award finalist poetry collection *Ghost Of* (2018). In his first foray into fiction, Vuong raises a fundamental question: "When does a war end?" (Vuong 2019b, p. 12). Rather than offering a simplistic or straightforward response, Vuong's narrative delves deep into the intricate dynamics that arise in the aftermath of trauma and displacement, whilst also imagining potential venues for survival and resistance for refugee subjects. Written as a letter to his illiterate mother, the novel features a Vietnamese American queer protagonist, Little Dog, whose circumstances are highly suggestive of Vuong's own: born in Vietnam in 1988, he flees the country with his family and arrives in the U.S. at the age of two. The family of refugees resettles in the working-class town of Hartford, Connecticut, where Little Dog is raised in poverty by his PTSD-stricken mother and schizophrenic grandmother. It is precisely these women's incessant stories about the war that Little Dog unearths throughout

his narrative as a survival strategy. In this way, the mother, Rose, and the grandmother, Lan, become the heart of the novel—by re-tracing their experiences and relationships, even if fraught and difficult, Little Dog addresses one of the novel's paramount concerns: how can we heal and endure the ripple effects of trauma, while envisioning more empowering possibilities for the conflict's *exit wounds*, the losses it leaves behind?

For this purpose, *On Earth*'s protagonist devises an intricate tapestry of magnified recollections, spanning his life and beyond—from his childhood and adolescence in Hartford, to his mother's and grandmother's experiences in Vietnam; the novel then reaches forward to the boy's future, when he relocates to New York to pursue his writing career. As a result, *On Earth*'s non-linear narrative, continuously shifting between the past and present, engages with a highly fragmentary form of temporal disorientation—one that I examine through the lens of queer temporalities. As Sara Ahmed (2006, p. 161) proposes, "[t]o make things queer is certainly to disturb the order of things [. . .] certain forms of living—certain times, spaces, and directions"—which have been ingrained in Western culture. Conventional notions of a subject's futurity, for instance, have traditionally revolved around "reaching certain points along a life course", which cumulatively create "the impression of a straight line" (Ahmed 2006, p. 554). To conform to this linear trajectory means abiding by what J. Halberstam (2005, p. 5) understands as "straight time": the prevailing logics of temporality in Western culture, which are governed by temporal regulations related to the "normative scheduling of daily life" around procreative family structures. Conversely, Halberstam underscores the notion of "queer time" as a form of temporality that emerges "in opposition to the institutions of family, heterosexuality, and reproduction", a concept that may thus encompass unconventional life trajectories that deviate from the established benchmarks of experience such as "birth, marriage, reproduction, and death" (pp. 1–3). The potentiality of queerness, however, extends here beyond sexual identity; it primarily lies in inspiring new life narratives and unconventional relationships with time. Being "an outcome of strange temporalities, imaginative life schedules, and eccentric economic practices" (p. 1), queer relations to time operate as points of resistance to normative temporal norms by challenging the principles on which these are firmly grounded: those of lineal progression, causality, reproduction and sequence.

As an illustrative mechanism enabling the workings of queer temporalities, Elizabeth Freeman (2010, p. 64) theorises what she terms the "temporal drag": a "distorting pull backward" in time, which both resists the forward trajectory of dominant conceptions of history and fosters a transformative dialogue between the past and the present. Yet, the most radical intervention enabled by the temporal drag lies in its ability to draw the past into the present as a means to uncover "what has been forgotten, abandoned, discredited, or otherwise effaced" (p. xiii). This intention aligns precisely with Vuong's purpose in *On Earth*: Little Dog's narrative present continuously encounters a pull backward in time, driven both by his own personal memories and the shattered retellings of the war by his mother and grandmother. Consequently, the temporal dimensions governing the novel transcend conventional boundaries between past and present and thus break free from the confines of linearity and sequence.

This article examines how such an alternative configuration of time serves a dual purpose for Vuong. By establishing these asynchronous, non-linear connections across time, Vuong brings to the forefront the silenced voices of the Vietnam War's *exit wounds*, thus contesting dominant accounts of the conflict—those embedded within U.S. imperialism. As Yến Lê Espiritu (2014, p. 20) contends, the production of official discourses on the Vietnam War and the construction of refugee subjectivities in the U.S. have been predicated on "mutually constituted processes of remembering and forgetting", which have historically relegated Vietnamese refugees into oblivion. In contrast, Espiritu proposes showing "that Vietnamese refugees are 'intentionalized beings' who enact their hopes, beliefs, and politics, even when they live militarized lives", thus foregrounding their "complex personhood" (p. 14). Evoking Espiritu's predicament, Vuong himself warns against the portrayal of the refugee as a "passive, needful, and pandering subject" (Vuong 2022). Contrarily,

Vuong intends "to reorient how we see refugees as people who are incredibly creative and innovative" (Vuong 2022).

On the other hand, *On Earth's* temporal configuration simultaneously disrupts the teleological narrative of progressive assimilation that is prevalent in refugee discourse by revealing the enduring forms of violence and precariousness that refugee subjects must face in contemporary America. Precisely, Vuong exposes the exceptionality of the "good refugee" story—one that is "almost always coded through upward mobility and economic success"—by demonstrating how refugeehood persists well beyond the subject's moment of resettlement (V. Nguyen 2019, p. 127). In doing so, Vuong rethinks the refugee category and contests "conventional understandings that confine refugee to a legal definition, short time frame, and pitiful existence" (V. Nguyen 2019, p. 111). In essence, by queering the normative temporality of the refugee experience, the novel envisions another way of "living historically" (Freeman 2010, p. xvi), showcasing how the characters' refugeehood is not finite but ongoing, necessitating a continuous search for healing and resilience.

## 2. On Remembering War

"[A]ll wars are fought twice, the first time on the battlefield, the second time in memory", V. T. Nguyen (2016, p. 4) declares. In his exploration of the production of hegemonic historical narratives, Nguyen argues that memories "are not simply images we experience as individuals"; rather, they are collected and collective, serving as "mass-produced fantasies" that contribute to the construction of prevailing historical discourses (p. 15). In this endeavour, memory becomes "a strategic resource in the struggle for power"; just as economic inequalities exist among different nation states and populations, there are also disparities in the construction of their collective memories (p. 10). As for the Vietnam War, Nguyen contends that despite the United States experiencing military defeat, "it won the war in memory on most of the world's cultural fronts outside of Vietnam", largely owing to its dominant position in industries such as filmmaking, video game production, book publishing, fine arts and historical archives (15). The American memory industry is, in fact, "[b]y far the most powerful of its kind" and serves multiple purposes: whilst it justifies and glorifies war, it likewise reinforces American national identity, exceptionalism and imperialism (p. 108).

In his dissection of dominant war narratives, Nguyen particularly criticises "disremembering": whilst admitting that some forgetting is inevitable, he opposes the kind of forgetting that "is not accidental but deliberate, strategic, even malicious", heralding a specific national narrative (p. 40). Illustratively, a prominent account of the Vietnam War stages "a US tragedy featuring US heroes and antiheroes, a blockbuster in which Southeast Asian play the supporting role", thus purposefully erasing the complexities of Vietnamese experience during and after the war (V. T. Nguyen 2012, p. 913). Nguyen likewise observes that the ethical approach of "remembering others" that emerged from the global antiwar movement proves to be similarly dissatisfying: this perspective tended to idealise and cast the Vietnamese as victims, perpetuating their portrayal as eternally suffering (V. T. Nguyen 2016, p. 77). Consequently, whether portrayed as dehumanised enemies or powerless victims, the Vietnamese have rarely been afforded full subjectivity within dominant Western narratives of the war. Nguyen contends that these are precisely the narratives in need of revision—an endeavour to which Vuong is surely dedicated. More specifically, Nguyen proposes the creation of a "just memory" as a potential antidote, a strategy that involves both reworking the past and confronting the present, "for it is today's material inequalities that help to shape mnemonic inequities" (p. 16). Firstly, a "just memory" provides a more complex understanding of human identity by recognizing simultaneously the humanity and inhumanity in ourselves and in others. Secondly, it gives equal access to the industries of memory to those who have been usually forgotten in dominant, nationalist war memories: "the weak, the poor, the marginalized, the different, and the demonized, or their advocates" (p. 18).

As an echo of Nguyen's formulations, it is certainly a *just memory* that Vuong intends to assemble in *On Earth*: whilst he foregrounds the inherent contradictions in human experience, he seizes the industry of memory itself by recovering the marginalised voices of Vietnamese refugees—Little Dog's and his family's. Countering the spectacularisation of refugee representations in the U.S. mainstream media, Little Dog's grappling with the past occurs through the fabric of the everyday—e.g., the quotidian storytelling sessions with his grandmother, Lan. In this vein, L. T. Nguyen (2020, p. 230) asserts that the way in which many refugee descendants learn about their family history is far from an "origin story passed down in full speech"; rather, parents and relatives will share memories in fleeting and antimonumental moments, sometimes "in the middle of watching a film or eating a meal". As Vuong states in a conversation with Viet Thanh Nguyen himself, he understands his writing as an "act of collaboration" that includes not only himself but also his family: "[t]here's one name on the book, but it's everyone there" (Vuong 2019a). In this spirit, Little Dog's mnemonic archive is designed through an active form of memory making, revealing how memory is not only constructed and incomplete, but also dialogic and intersubjective. Most importantly, by amplifying Lan's individual story, Little Dog is giving an account of the violence that is experienced not only by forcibly displaced populations, but especially by the Vietnamese women who were multiply disadvantaged during the war—by class, ethnicity, gender and different nation-states' power.

In particular, the novel challenges dominant discourses on the Vietnam War by deploying what Freeman (2010, p. 64) terms the "temporal drag": a "distorting pull backward" which disrupts "the dominant arrangement of time and history" wherein temporal parameters are arranged "into consequential sequence" (p. xi). Instead, the temporal drag attunes to the "ways that nonsequential forms of time" can unveil historical material "that may be invisible to the historicist eye" (p. xi). In *On Earth*, Little Dog and his family's narrative past is constantly dragged forward into their present through the storytelling practices of the grandmother, Lan—a war victim and a refugee. During these quotidian and antimonumental moments, Lan's stories range from traditional Vietnamese "mythologies of manlike monkeys" to harsher "[s]cenes from the war" (Vuong 2019b, p. 22), whilst tracing more personal and intimate accounts: we learn, illustratively, that Little Dog's mother was born "of the white American serviceman deployed on a navy destroyer" (p. 23). In these scenes of cross-temporal exchange, Lan's fractured diasporic memory works as a "*productive* obstacle" to the supposedly progressive trajectory of normative history and offers an alternative to hegemonic historical accounts (Freeman 2010, p. 64).

The workings of the temporal drag, however, do not propose a restoration of an idealised past. As Freeman (2010, p. 7) suggests, time appears to "'bind' history's wounds" through corporeal sensations, as certain bodies register "on their very surface the co-presence of several historically contingent events" (p. 63). In Lan's case, the enduring impact of past trauma is ubiquitous: "Lan's back was perpetually bent"; "her nails", impeccably manicured by her daughter, "were the only unblemished thing about her" (Vuong 2019b, pp. 17–19). Indeed, the traces of historical violence are deeply etched in both the grandmother's physical being and psyche. As a result, Vuong dwells on Lan's memories to explore what "living with injury" means (Love 2007, p. 4), both physically and metaphorically: as Heather Love (2007, p. 1) cautions, "[f]or groups constituted by historical injury, the challenge is to engage with the past without being destroyed by it". In order to overcome this obstacle, Vuong's evocation of the past reflects a twofold focus on trauma and healing, as he revisits memories of both violence and resilience. Vuong's novel, hence, foregrounds the complexities that are inscribed in the refugee experience, thus questioning the fixity of history, memory and identity.

This ambivalent dragging of "history's wounds" is epitomised in *On Earth* in the episode recounting the story of a "woman, a girl, a gun" located in Part I—a memory that Little Dog has assumedly inherited during one of Lan's storytelling sessions (Vuong 2019b, p. 35). Juxtaposing two scenes of brutality set in Vietnam during the war, this episode interrogates the possibilities of envisioning a "just memory", one that attempts to remember

"what might be forgotten, accidentally or deliberately, through self-serving interests, [or] the debilitating effects of trauma" (V. T. Nguyen 2016, p. 17). The chapter begins with a "woman waiting on the shoulder of a dirt road, an infant girl wrapped in a sky-blue shawl in her arms", detained at a checkpoint as she watches a Huey helicopter burn her village to the ground (Vuong 2019b, p. 35). Soon enough, a sobering realisation dawns upon us: the mother and daughter are twenty-eight-year-old Lan and Little Dog's newborn mother, Rose. Introducing a parallel vignette, the narrative switches abruptly to "the back of a garage lit with a row of fluorescent lights", where "five [soldiers] have gathered around a table" to drink (p. 36). As the scene unfolds, one of the men leads into the room a leashed dog-sized monkey, a macaque, who has been "force-fed vodka and morphine in its cage all morning" (p. 38). In what follows, the performance of a gruesome ritual unfolds: the men "cut open the macaque's skull" and "take turns consuming the brain, dipped in alcohol [...] while the monkey kicks beneath them" (p. 41). As Little Dog explains, this meal was believed to rid men of impotence during the war; the soldiers, hence, devour it for "the future of their genes" (p. 41).

Such a justification for this horrid practice is remarkably suggestive of Halberstam's "straight time", whereby the heterosexual temporal schemes of reproduction and inheritance connect the family not only to "the historical past of the nation", but also "to the future of both familial and national stability" (Halberstam 2005, p. 5). Offering a telling reflection on the workings of memory and the production of nationalist war narratives, this episode shows how, for the soldiers, providing a future—both familial and national—requires erasing the victims' experience of the war in favour of a dominant, coherent memory that glorifies their military intervention. More precisely, Vuong describes how the macaque's brain is "the closest, of any mammal, to a human's": these animals "are able to recall past images and apply them to current problem solving"; macaques, this means, "employ memory in order to survive" (Vuong 2019b, pp. 41–43). Consequently, the soldiers' violent consumption of the animal's brain entails that "all of its memories dissolve into the men's bloodstreams", hence annihilating any possibility for its survival (p. 43). Ultimately, the monkey thus becomes a brutal metaphor for the conflict's victims at large, whose memories have been deliberately disremembered by dominant narratives: "[a] story, after all, is a kind of swallowing", Little Dog declares (p. 43).

As an opposing exercise to the soldiers' mnemonic "swallowing", Little Dog is determined to foreground survival in this chapter through the juxtaposed recuperation of his grandmother's past recollections: avoiding freezing Lan into her victimhood, the boy focuses on her capacity for resilience, her agency, especially symbolised in the act of naming. The narrative switches back to young Lan, who, detained at the checkpoint, notices one of the soldiers' name stitched on his chest, "[t]he possession of a name, after all, being all they share" (p. 39). Whilst this observation recognises the enemy's paradoxical humanity and inhumanity—an essential precondition for envisioning a *just memory*—it also reminds Lan of the story behind her own name. Having been "born nameless", it was not until she was seventeen that she named herself *Lan*—meaning *orchid* in Vietnamese—after escaping "from her arranged marriage to a man three times her age" (p. 39). Disowned by her family, Lan resorted to sex work to sustain herself and her daughter during the war: "it was her body, her purple dress, that kept her alive", Little Dog explains (p. 23). The night that her village is razed, "Lan stares into the dark, thinking of another world", a more hospitable futurity where she tells her child "the story of a girl who ran away from her faceless youth only to name herself after a flower that opens like something torn apart" (pp. 40–41). Storytelling, indeed, becomes the channel through which Lan's legacy, her spirit of endurance, is passed on both to her daughter and grandson. Whereas Lan's defining act of survival registers through her self-naming, writing herself into existence, Little Dog's resides in the recuperation of his grandmother's voice, dragging her story into our present and juxtaposing it to the American memory industry's violent swallowing—survival becomes a creative force in both instances. Yet, Little Dog's creative force in the novel does not only unfold

through his treatment of the past, but also through his exploration of ongoing experiences of refugeehood. It is to this analysis that I now turn.

## 3. Queering Refugeehood

Returning to the connection between refugee survival and the issue of timing, V. Nguyen (2019, p. 109) raises a pivotal question: "When does a refugee stop being a refugee?". The question of temporality, or "the duration of the refugee category", lies at the heart of contemporary discourses on refugee production; it is in this regard that V. Nguyen (2019, pp. 110–12) advocates thinking of refugeehood as "an enduring creative force [...] connec[ting] past, present, and future forms of displacement"—a concern that permeates *On Earth.* The novel's narrator retraces the experiences and relationships in which he engages while growing up in Hartford, Connecticut, to unveil the various forms of violence that shape his life after resettlement. Crucially, Little Dog's and his family's diasporic experience is deeply conditioned by their complex identity as refugees: as Amy Kaplan (2002, p. 1) contends, "international struggles for domination abroad profoundly shape representations of American national identity at home, and [...] in turn, cultural phenomena we think of as domestic or particularly national are forged in a crucible of foreign relations".

On this matter, M. Nguyen (2018, p. 135) sheds light on how the aftermath of the Vietnam War compelled the U.S. to desperately seek to re-build its political and military status; this was attempted by constructing a discourse of the refugee not as a strictly legal category but rather "as a 'moral-political tactic'". In the months that followed the Fall of Saigon in 1975, the American media became saturated with images of mass migration depicting "racialized dramas of powerless, vulnerable Vietnamese void of their own varied decolonizing politics" (p. 135). These portrayals had a great impact on popular consciousness and helped craft a new narrative that reaffirmed American exceptionalism, reinforcing the U.S. noble, ethical actions towards Vietnamese refugees: "[b]y 'saving' refugees, the nation refashions its role as agent of war to humanitarian leader", consequently erasing the complex subjectivities and realities of forcibly displaced people and eliding the geopolitical circumstances that produced them (p. 136). On a similar note, Espiritu (2014, p. 4) argues that the most popular narratives of the Vietnam War continue to depict Vietnamese refugees as "incapacitated objects of rescue" in need of care—care that is presumably best administered by and within the U.S. By doing so, these narratives formulate the refugee not only as a problem to be resolved, but also as a solution that is "deployed to rescue the Vietnam War for Americans" (p. 83).

Furthermore, the predominant refugee rhetoric that emerged in the wake of the war was also predicated on a telos of progression that urged the voluntary "Americanization of displaced people": refugees were expected to rapidly re-narrate as immigrants to secure their opportunity of becoming American citizens (M. Nguyen 2018, p. 136). In this vein, M. Nguyen (2018, p. 127) notes how these pressures for refugee assimilation—including religious conversion and adoption of the English language—placed refugees in a position of indebtedness towards their resettlement country, which required their relentless expression of gratitude on an everyday level. Nguyen's formulations resonate strongly with Hannah Arendt's essay "We Refugees", in which she theorises refugee subjectivity as demanding performance to satisfy the host country's expectations. Arendt (1994, p. 110) argues that even if "refugee" is a necessary category that provides displaced subjects with legal recognition, it simultaneously becomes an identity that they must strive to leave behind due to the resettlement country's assimilationist expectations. Much like Nguyen, Arendt (1994, p. 117) argues that refugees must "prove [their] loyalty" under the constant threat of exclusion, even if this means being "complicit in the public erasure of their own grievances and histories" (M. Nguyen 2018, p. 129). As a result, refugees are placed at an impossible crossroads by American assimilationist ideology, where they must choose between "extracting [or] maintaining one's roots", a position that evidently causes immense emotional devastation and fragmentation in the surviving subjects (M. Nguyen

2018, p. 129). As a de-historicizing device, assimilation therefore crafts fixed notions of identity that disavow the political, cultural and historical complexities that are inherent to experiences of war and displacement, as well as their "temporal and spatial multivalence" (M. Nguyen 2018, p. 129).

In opposition to this teleological configuration of refugee subjectivity—one that certainly complies with the logics of *straight time*—the refugee experiences portrayed in *On Earth* eschew any definition of refugeehood as a finite, temporary condition by foregrounding the multiple forms of violence that displaced subjects must face after resettlement. V. Nguyen (2019, pp. 113–14) addresses this issue by arguing that the refugee experience is far from brief or temporary; instead, "most refugees in the world experience their condition as refugees indefinitely, sometimes for an entire lifetime"—a stark reality for Little Dog and his family. Specifically, the porous temporalities that permeate the experiences of Little Dog, Rose and Lan convey a queering of refugeehood, a reconfiguration of the refugee identity as a continual negotiation between submitting to assimilationist imperatives and reclaiming their own histories, all the while imagining possibilities for healing in the aftermath of trauma. Refugeehood, thus, continues to shape the contours of their diasporic lives, even long after their arrival in the U.S. In this sense, Vuong's queering of refugeehood invokes the coming into consciousness that V. Nguyen (2019, p. 110) terms "refugeetude": a "critical reorientation, an epistemological shift in how we think about and understand the category refugee". As analysed in what follows, *On Earth* employs this critical reconfiguration to expose "connections to past, present, and future forms of displacement" impacting refugee subjects (p. 110): whereas Rose's experience reveals her exploited condition within the current neoliberal economy, Little Dog's refugeehood is intricately interwoven with his queerness, rendering it susceptible to various forms of contemporary homophobic violence.

*3.1. No Refuge: Trauma and Neoliberal Exploitation*

In *On Earth*, the inherent tensions in the experience of refugeehood find a nuanced portrayal through the character of Rose, Little Dog's mother, who is featured in the myriad flashbacks to the protagonist's childhood and adolescence. These recollections are often brief, chaotic and fragmented; such incoherent temporal parameters vividly illustrate the "peculiar, temporal structure, the belatedness" of trauma (Caruth 1996, p. 17). For Cathy Caruth, traumatic experiences, given their overwhelming and shocking nature, may initially be repressed, only to resurface unexpectedly later on. This return of the repressed often disrupts conventional narrative structures, challenging established notions of linearity and coherence. Roger Luckhurst (2008, p. 9) echoes a similar perspective, asserting that "no narrative of trauma can be told in a linear way; it has a time signature that must fracture conventional causality". Indeed, Rose's life in the U.S. is replete with instances of injury and trauma: "When does a war end? When can I say your name and have it mean only your name and not what you left behind?", Little Dog asks her (Vuong 2019b, p. 12). As we follow Little Dog through his formative years, we comprehend that "war was still inside [Rose], that once it enters you it never leaves" (p. 4). Rose's everyday life is persistently disrupted by the impact of PTSD, as exemplified by numerous incidents: Little Dog recalls "[t]hat time when I was five or six and, playing a prank, leapt out at you from behind the hallway door, shouting, 'Boom!'. You screamed, face raked and twisted, then burst into sobs" (Vuong 2019b, p. 4). Rose's experience, therefore, exemplifies the complexities of narrating trauma, wherein the temporal dissonance and delayed manifestation of traumatic events disrupt conventional narrative time. Besides the enduring remnants of past trauma saturating Rose's daily experience, Little Dog's flashbacks equally uncover the traces of another attritional form of violence inflicting his and his mother's ongoing refugeehood in the U.S. Resettled in a country whose precarious and racialised economic system turns refugee survival into an incommensurable struggle, V. Nguyen (2019, p. 113) states that refugees continue to be perceived as "figures of lack—homeless specters, abject outsiders, identityless mass, or wastes of globalization". Particularly, refuge in the U.S. is profoundly inscribed in capitalism, which, along with other

factors such as race and gender, binds refugee subjects to the current neoliberal economy that perpetuates their search for resettlement (p. 119). In other words, refugeehood does not end when subjects are granted political asylum; rather, this moment is just the beginning of "a life of low-wage labor, with few opportunities for upward mobility", a reality that is at odds with the prevailing discourse of refugee exceptionalism "whereby the refugee's struggle and suffering are cast as provisional" (p. 110). Placed in a position that is marked by labour exploitation, economic precarity and alienation, refugees are thus "reified as not quite human", their lives rendered both disposable and untenable in an ever-lasting process of unsettlement (p. 113).

These very structures of violence, manifested in acute forms of institutionalised "lacking", certainly continue to press upon Rose's identity as a refugee, a condition that is inevitably inflected by her gender and class. In one of Little Dog's vivid flashbacks, he contemplates the exploitative labour conditions that Rose endures over many years of strenuous, poorly compensated work as a manicurist at a nail salon, a job in which "there are no salaries, health care, or contracts, the body being the only material to work with and work from" (Vuong 2019b, p. 80). The nature of Rose's profession is significant. As Miliann Kang examines, the nail salon industry stands as one of the fastest-growing Asian immigrant-dominated businesses in the U.S. and holds a complex position within the Vietnamese American population, as it constitutes an ambivalent site of economic potential and labour exploitation, simultaneous resistance to and complicity with racial discrimination and structural violence. Additionally, this form of body service work is not only racially and economically stratified but also gendered, with working-class immigrant women constituting most of the workforce. Kang (2003, p. 824) explicates that shared factors among Vietnamese women, such as limited English language ability, unrecognized professional credentials from their country of origin and undocumented immigration status, contribute to the racialised perception of Asian women as "productive and docile workers, whose 'nimble fingers' make them desirable and exploitable" within the U.S. labour market, which is shaped by racial hierarchies. Rose's labour, therefore, illuminates the intersections of race, gender and class inequalities shaping the experiences of refugee women in the global service economy.

In connection to the precarities engendered by contemporary neoliberalism, Lauren Berlant (2011) eloquently notes that exhaustion—as well as its adjacent states such as fatigue and burnout—has become one of its primary affects. Widespread in the current climate of economic precarity, exhaustion contributes to what Berlant (2011, p. 104) terms "slow death", that is, the "destruction of life, bodies, imaginaries, and environments by and under contemporary regimes of capital". It is impoverished, working-class and marginalised subjects, being the sustaining force of the neoliberal economy, who experience slow deaths the most—refugees being no exception to this. Whereas exhaustion often operates as a normalised form of violence, taking place quietly off-scene, Vuong renders it visible by exposing its devastating effects in *On Earth*. Rose's physical condition—much like Lan's— serves as unmistakable evidence of her situation: her lungs are inflamed from exposure to harmful chemicals, her hands rough and blistered "like two partially scaled fish", her back crippled from bending over clients (Vuong 2019b, p. 79)—a posture that situates her in literal inferiority to her predominantly white, middle-class customers, further alienating her as a subject. Little Dog's documentation of his mother's wounded body, in fragments, exposes the eerie similarities between past and present histories of injury: it is not the American war machine that continues to inflict violence on Vietnamese bodies now, but its contemporary gendered and racialised economic structures. Echoing Nguyen's words, Little Dog interprets Rose's precarious condition as "the wreck and reckoning of a dream", a stark reminder of "what it means to be awake in American bones—with or without citizenship—aching, toxic, and underpaid" (pp. 80–81).

In the novel, the nail salon works as an immobilising force, an almost inescapable reality for Vietnamese women. As Little Dog recalls, many of Rose's fellow manicurists would hope "the salon would be a temporary stop": "*I won't stay here long*, we might say.

*I'll get a real job soon*" (p. 80). In most cases, however, their disposable jobs at the nail salon would become their permanent form of livelihood, even if utterly insufficient. This depiction therefore captures the overlooked experiences of Vietnamese women refugees, who, dealing with the absence of missing or dead family members, must support their families through this form of menial, low-wage, labour-intensive work. Hence, Little Dog's refugeetude—a making sense of his family's experience—here forefronts the "failure of the neoliberal nation-state to provide refugeed individuals like [Rose] a form of livable refuge" (V. Nguyen 2019, p. 121). By uncovering the condition of disposability and the protracted forms of harm that await Rose after fleeing Vietnam, Vuong's analysis of the experience of (a) refugeI in the U.S. is thus primarily framed through a critique of its neoliberal system's dehumanizing practices. Most importantly, Vuong's depiction reveals how these processes of neoliberal exploitation are not only inflected by race and class, but also gender, consequently reinforcing broader structures of inequality that situate Vietnamese refugee women in a position of multiple forms of discrimination. In this way, Rose's experience not only illustrates how the gendered labour that she engages in is a direct result of the war's impact, but also challenges the U.S. exceptionalist "good refugee" narrative, whereby refugees are expected to assimilate into American mainstream society by achieving economic success and upward mobility (V. Nguyen 2019).

### 3.2. Intersecting Realities: Refugeehood and Sexual Identity

Whilst the portrayal of Rose's refugeetude evidences the precariousness shaping her diasporic experience within contemporary capitalism, Little Dog's depiction similarly exposes present traces of violence complicating his survival after resettlement. In particular, the novel underscores how the protagonist's experience of refugeehood is intricately intertwined with his queer sexuality, his intersectional subjectivity thus situating him within multiple axes of power and difference. As the panoply of flashbacks to Little Dog's adolescence transpire, his brutally honest exploration of queerness is pivotally impacted by his relationship with Trevor, a white boy whom he meets at 14 during one of his low-paying jobs on a tobacco farm. In addition to being the grandson of the farm's owner, Trevor is battling demons of his own—an alcoholic and abusive father and an opioid addiction. The first time Little Dog sees him, he notices Trevor's "finely boned face dirtstreaked under a metal army helmet, tipped slightly backward, as if he had just walked out from one of Lan's stories" (Vuong 2019b, p. 94). Not only does this initial encounter recall the violence left behind by Little Dog's family, but it also foreshadows the strenuous nature of the relationship that is about to unfold. Indeed, the initial exchanges between the two boys are heavily punctuated by the protagonist's submission to Trevor's brusque sexual demands and raging homophobia: "You think you'll be really gay, like, forever?", Trevor asks him, "I think me…I'll be good in a few years" (p. 188). In this way, the uneven sexual dynamics operating between them point to both present discourses of sexuality and prior and ongoing histories of racism. Little Dog, however, is no stranger to abusive behaviour. Trevor's mistreatment evokes the repeated abuse that the protagonist endures at the hands of his mother, an experience he attributes to her PTSD from the war: "Violence was already mundane to me, was what I knew, ultimately, of love" (p. 119). Over time, the scenes of intimacy unfolding between Trevor and Little Dog foster a more respectful mode of relationality between the two. Their queer desire, nonetheless, is always only allowed to be explored temporarily: "there was a space for us: a farm, a field, a barn, a house, an hour, two" (p. 111). Beyond this momentary reprieve, Little Dog's multiply oppressed positionality subjects him to constant suffering: being taunted by his classmates with labels such as "freak, fairy, fag" (p. 14), discovering the tag "FAG4LIFE" scrawled across his front door (p. 180), having his hot-pink bicycle vandalized by a ruthless teenager—on that day, he confesses, "I learned how dangerous a color can be" (p. 134).

Another pivotal juncture shaping Little Dog's maturation, further unravelling the intersections between his queerness and refugeehood, occurs when he discloses his sexual orientation to his mother in a Dunkin' Donuts cafeteria—an ordinary setting starkly

contrasting with the extraordinary news that he is about to deliver. "I don't like girls", he straightforwardly confesses to Rose (p. 130). His choice of words is deliberate, refraining from using the Vietnamese word for homosexual: "pê-đê—from the French pédé, short for pedophile. Before the French occupation, our Vietnamese did not have a name for queer bodies—because they were seen, like all bodies, fleshed and of one source", he contextualizes (p. 130). Language thus becomes a poignant link between past and present homophobic and imperialist violence, a legacy that Little Dog consciously rejects as an assertion of his agency. Rose's immediate response is one of fear and protectiveness: "They'll kill you", she blurts out, "you know that" (p. 130). The mother's words echo the persistent warning that she tirelessly tries to convey to her son throughout his childhood, now seemingly in vain: "don't draw attention to yourself. You're already Vietnamese" (p. 219). Rose, acutely cognizant of the social violence that is inflicted on those who are multiply disadvantaged, seeks to underscore this awareness by unloading, to her son's surprise, a painful truth of her own: back in Vietnam, at seventeen, she was compelled by her husband to undergo an abortion, a decision attributed to the family's impoverished situation. This parallel act of disclosure reinforces the urgent warning that Rose imparts to her son, reminding him how their bodies are both afflicted by multiple vulnerabilities, stemming not only from their refugeed condition, but also from their gender and sexuality. Despite her initial reaction, Rose refuses to lose another son: "You don't have to go anywhere. It's just you and me, Little Dog. I don't have anyone else" (p. 131). Little Dog, however, recognizes the underlying truth in Rose's words: a few lines later, he ruminates on two homophobic crimes that were committed merely months before his coming out, one in Vietnam, the other in the U.S. Hence, far from marking a conclusion to his realization of a queer identity, this momentous encounter evidences how Little Dog's refugeetude entails an ongoing negotiation, grappling with the complexities that are inherent to his identity throughout his lifetime. Ultimately, the protagonist's survival in contemporary America remains continually threatened not only by his refugeehood but also by his queerness.

## 4. Conclusions

On the very first pages of *On Earth We're Briefly Gorgeous*, Little Dog ruminates on the migration of monarch butterflies "from southern Canada and the United States to portions of central Mexico, where they will spend the winter" (Vuong 2019b, p. 4). As we read on, this form of animal imagery becomes a recurrent device framing the text's swirling detours—an allegory for both Little Dog's personal diasporic journey and the novel's temporal dislocation. Beginning in autumn, the boy elucidates how the colony's expedition is triggered by time, "by the angle of sunlight, indicating a change in season, temperature, plant life, and food supply" (p. 8). Butterflies, therefore, are set in motion by their timely instincts of survival: "[i]t only takes a single night of frost to kill of a generation. To live, then, is a matter of time, of timing" (p. 4). The migratory journey that brings Little Dog and his family to the U.S. is likewise "a matter of time", precipitated by the need to flee and survive—not winter, but the Vietnam War's aftermath. Yet, the family's past is not simply left behind. Unsettling conventional temporal boundaries through the deployment of queer temporalities, Little Dog drags his family's past into the present, giving voice to his grandmother—a war victim and a refugee—and thus devising a more inclusive, *just memory* of the war that accounts for the complexity of the refugee experience. In so doing, Vuong's rendition of refugeehood is reconfigured as an "enduring creative force", which not only "opens up new ways of conceptualizing refugee subjects", but also establishes "connections to past, present, and future forms of displacement" (V. Nguyen 2019, pp. 110–11).

In this critical reorientation of how experiences of refuge(es) are represented—what Vinh Nguyen terms "refugeetude"—Vuong likewise foregrounds the porous temporalities in which these are imbued. Vuong, in other words, queers the conception of the refugee as a temporal and temporary category, demonstrating how refugeehood is not finite but ongoing for both Little Dog and his family: it is not the American war machine that continues

to inflict violence on them in the present, but their precarious condition as disposable, dehumanised and exploited subjects in contemporary America. Nonetheless, responding to V. Nguyen's (2019, p. 111) call for "theorizing how refugeeness might engender modes of perceiving, critiquing, and resisting the very structures of violence that produce and continue to press upon refugees", Vuong avoids reducing Little Dog, Rose and Lan to their perennial victimhood. Instead, he underscores their complex subjectivities, redirecting our understandings of refugeehood from "a crisis to be resolved" to "an experiential resource for developing significant and durable ways of being in and moving through the world" (V. Nguyen 2019, p. 111). By introducing new conceptualisations of refugee subjects, Vuong illustrates how the identities of *On Earth*'s characters as refugees are not fixed or transitory but indefinite and lingering, necessitating a continuous navigation between violence and creation. Essentially, this illuminates how their identities, too, are "always in temporal drag", their present perpetually being intertwined with their past (Freeman 2010, p. 93). Such a dynamic and resilient consciousness becomes pivotal for the novel's protagonist, whose intersectional identity as a queer refugee situates him within multiple axes of power. As a direct product of war, he must simultaneously negotiate between trauma and healing. In the poem "Notebook Fragments" from his collection *Night Sky with Exit Wounds*, Vuong encapsulates the paradox of his existence: "An American soldier fucked a Vietnamese farmgirl. Thus my mother exists. Thus I exist./Thus no bombs = no family = no me" (Vuong 2016, p. 70). Little Dog—and, by extension, Vuong—becomes thus an *exit wound* himself, a living testament to the conflict's multifaceted and often conflicting repercussions, as well as the inherent tensions in the refugee experience. Vuong's ultimate emphasis, however, is always on the willingness to survive, the possibility of imagining a more liveable future, even if only briefly.

**Funding:** This research has been possible thanks to the grant FPU20/07148, funded by MCIN/AEI/10.13039/501100011033 and "FSE invierte en tu futuro".

**Data Availability Statement:** No new data were created or analyzed in this study. Data sharing is not applicable to this article.

**Conflicts of Interest:** The author declares no conflict of interest.

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
