# Peer review of "“To Live Is a Matter of Time”: Memory, Survival and Queer Refugeehood in Ocean Vuong’s On Earth We’re Briefly Gorgeous"

_humanities, doi:10.3390/h13020041_

Round 1
Reviewer 1 Report
Comments and Suggestions for Authors
This proposal – elegantly written – presents a very cogent analysis and discussion of the novel On Earth We’re Briefly Gorgeous by Ocean Vuong. I think the manuscript constitutes an innovative contribution to the discussion of the topics of war, human forced mobility, memory and mediatic and literary representation of the Vietnam War, not only in the US American collective public imaginary, but also on a global scale. The manuscript masterfully integrates up-to-date critical sources related to the concept of refugeehood and temporality with the interpretation of Vuong’s fiction and critical work. It offers an insightful discussion of the dissenting and revelatory functions that the use of queer temporalities can have to represent refugee stories and uncover silenced stories. The manuscript demonstrates the potential that concepts such as Vuong’s refugeetude can have to challenge misleading and dehumanising representations of refugees. All in all, I think the manuscript should be accepted in its present form.
Author Response
RESPONSE TO REVIEWER 1:
Thank you very much for taking the time to review this manuscript. Please find the corresponding revisions in track changes in the re-submitted file.

Reviewer 2 Report
Comments and Suggestions for Authors
Please, look at the comments in the file.
In general, this is a very interesting research paper as it presents a thorough analysis of intersectionality in such a particular identity and experience. Maybe more about the queerness of the protagonist should have been included or commented and linked to the queer theory exposed.

Author Response
RESPONSE TO REVIEWER 2:
- Summary:
Thank you very much for taking the time to review this manuscript. Please find the detailed responses below and the corresponding revisions in track changes in the re-submitted file.
- Point-by-point response to Comments and Suggestions for Authors
Comment 1: will this have any importance further in the article?
Comment 2: this section is interesting but needs more clarity when discussing dragging the wounds to certain topics (for instance, soldier's violence when eating the monkey's brain, or Lan's past)
Response to Comments 1 and 2: In order to emphasize the importance of the "temporal drag" in the novel's analysis (as a “productive obstacle” to the teleological arrangement of history) and to ensure a clearer discussion of its connection to the specific scene under examination (involving the lingering impact of past war scenes depicting both the soldiers' violence and Lan's experience), this section has undergone revision and restructuring. Please refer to the updated manuscript for further details.
Comment 3: why the first name here and not before?
Response to Comment 3: I provide Marguerite Nguyen's complete name on this occasion since this marks the initial citation of her in this text. Previous mentions of "Nguyen" in this text pertain to Viet Thanh Nguyen, Vinh Nguyen, or Ly Thuy Nguyen. For each author's first citation or instances where ambiguity could arise, I consistently include their first names.
Comment 4: here, the same
Response to Comment 4: I have erased Yen Le Espiritu’s first name here because this is not the first time I cite her in the text.
Comment 5: it would also be interesting to see some more analysis related to the queer nature of the protagonists, and offer a more intersectional approach to his experience
Comment 6: Maybe more things can be commented on the protagonist's own queerness
Response to Comments 5 and 6: I concur with the reviewer's observation that the article requires a more in-depth analysis of the protagonist's queerness. To address this concern, I have introduced a new section ("3.2. Intersecting Realities: Refugeehood and Sexual Identity") specifically focusing on this aspect. To adhere to the word count limit, I had to eliminate the previous section 3.2 and replace it with the new one, aiming to offer a more pertinent analysis within the article's scope.

Reviewer 3 Report
Comments and Suggestions for Authors
In general, the close readings of the text are good but sometimes gets lost when author is explaining the theories. I think there is too much reliance on other people’s theories. It’s hard to hear the author’s voice in this essay, and how author can use others’ theories to enhance the author’s argument, rather than to be led by them. In this way, there is both too much explanation of others’ theories and not enough strategically. Sometimes, there are lengthy discussions and uses of someone else’s theories at the cost of more close readings of the text led by the author. I would go back and try to reorganize but only after the author can reconcile how healing of trauma and giving agency to the refugee is anything other than saying just that without using the theories of queer temporalities. It also feels like the theories author is using have their own separate discussions. The argument that the author presents in the beginning of when does war end and how can we heal trauma and leave behind the losses (line 40) contradict what author is saying especially in the discussion of refugee as a continuing and permanent condition. There is clarity with the theories about refugees (missing some vectors like gender and sexuality) but that and the queer reading of the novel seem to be two different papers. How do these come together into a coherent theory and into a reading of the novel?
It is also difficult understanding some of the theories on queer temporalities. How does this essay engage a queer’d reading of the novel by instantiating refugee remembering as opposed to the dominant narrative? In many of theories author uses, particularly temporal drag, I am not sure how this is different from just saying, Vuong weaves together past and present and reformulates a creative “productive obstacle” to dominant cultural narratives about the refugee. In this sense, how does queering a reading of the novel and its temporal drag work differently? What is specific or appropriate about using temporal drag as opposed to other ways of understanding this text? For example, in lines 172-215: These two paragraphs need more work in explaining what temporal drag is. It seems that there is a dragging of past into the present that then is etched into the body? It’s not clear how this is working or how this theory serves this reading. Isn’t this what people do? Do you mean active rather than seeing refugees as passive?
What about gender and sexuality? This seems really important and is missing in this reading. See line 379 where author mentions via Nguyen’s quote about race and gender. So far, there is no discussion of the gendered nature of this experience even as author speaks about the experiences of Lan and Rose. This seems particularly important in a discussion of the novel’s refugee experience. Lan has a mixed raced child, Rose ( I believe she is. I could be misremembering, but definitely Little Dog is mixed) impregnated by the white enemy. Lan also must engage in sex work to survive. How does a queer’d reading of this novel then not engage with gendered and sexual aspects of their identities, including Little Dog’s? Much of the discussion of US imperialism and the neoliberal assimilation of the refugee toward the end needs to engage with the gendered nature of this war, and its effects on Vietnamese women.
Try making sure your main points are front loaded, so at the top of the paragraph. Often, paragraph ideas appear near or at the end.
Author Response
RESPONSE TO REVIEWER 3:
- Summary
Thank you very much for taking the time to review this manuscript. Please find the detailed responses below and the corresponding revisions in track changes in the re-submitted file.
- Response to the reviewer’s comment:
Comment:
In general, the close readings of the text are good but sometimes gets lost when author is explaining the theories. I think there is too much reliance on other people’s theories. It’s hard to hear the author’s voice in this essay, and how author can use others’ theories to enhance the author’s argument, rather than to be led by them. In this way, there is both too much explanation of others’ theories and not enough strategically. Sometimes, there are lengthy discussions and uses of someone else’s theories at the cost of more close readings of the text led by the author. I would go back and try to reorganize but only after the author can reconcile how healing of trauma and giving agency to the refugee is anything other than saying just that without using the theories of queer temporalities. It also feels like the theories author is using have their own separate discussions. The argument that the author presents in the beginning of when does war end and how can we heal trauma and leave behind the losses (line 40) contradict what author is saying especially in the discussion of refugee as a continuing and permanent condition. There is clarity with the theories about refugees (missing some vectors like gender and sexuality) but that and the queer reading of the novel seem to be two different papers. How do these come together into a coherent theory and into a reading of the novel?
It is also difficult understanding some of the theories on queer temporalities. How does this essay engage a queer’d reading of the novel by instantiating refugee remembering as opposed to the dominant narrative? In many of theories author uses, particularly temporal drag, I am not sure how this is different from just saying, Vuong weaves together past and present and reformulates a creative “productive obstacle” to dominant cultural narratives about the refugee. In this sense, how does queering a reading of the novel and its temporal drag work differently? What is specific or appropriate about using temporal drag as opposed to other ways of understanding this text? For example, in lines 172-215: These two paragraphs need more work in explaining what temporal drag is. It seems that there is a dragging of past into the present that then is etched into the body? It’s not clear how this is working or how this theory serves this reading. Isn’t this what people do? Do you mean active rather than seeing refugees as passive?
What about gender and sexuality? This seems really important and is missing in this reading. See line 379 where author mentions via Nguyen’s quote about race and gender. So far, there is no discussion of the gendered nature of this experience even as author speaks about the experiences of Lan and Rose. This seems particularly important in a discussion of the novel’s refugee experience. Lan has a mixed raced child, Rose ( I believe she is. I could be misremembering, but definitely Little Dog is mixed) impregnated by the white enemy. Lan also must engage in sex work to survive. How does a queer’d reading of this novel then not engage with gendered and sexual aspects of their identities, including Little Dog’s? Much of the discussion of US imperialism and the neoliberal assimilation of the refugee toward the end needs to engage with the gendered nature of this war, and its effects on Vietnamese women.
Try making sure your main points are front loaded, so at the top of the paragraph. Often, paragraph ideas appear near or at the end.
Response to the reviewer’s comment:
As per the reviewer's suggestions, various paragraphs within the article have undergone revisions and rearrangements to improve the clarity and coherence of the main arguments. Please refer to the updated manuscript for further details.
Regarding the section on queer temporalities and its application in the analysis of the text, Section "2. On Remembering War" has been carefully reorganized, particularly the concluding part, to elucidate how the mechanism of temporal drag is employed in interpreting the novel.
Acknowledging the reviewer's valid point about the need for a more in-depth exploration of gender and sexuality, several adjustments have been implemented:
- In section "2. On Remembering War," a clarification has been incorporated regarding Lan's gendered experience of violence during the war. Due to constraints in word count, a more detailed analysis could not be provided, but efforts have been made to convey the essence.
- Section "3.1. No Refuge: Trauma and Neoliberal Exploitation" now includes an analysis of how Rose's experience of exploitation within contemporary capitalism is influenced not only by her class and refugee status but also by her gender.
- A new comprehensive section ("3.2. Intersecting Realities: Refugeehood and Sexual Identity") has been introduced to address the intersections between the protagonist's refugeehood and his queerness. To accommodate this addition while adhering to the word count limit, the previous Section 3.2 has been replaced with the new one, ensuring a more relevant and focused analysis within the article's scope.

Round 2
Reviewer 3 Report
Comments and Suggestions for Authors
I just have a few edits and comments:
118-19: the sentence doesn’t quite make sense, “much like economic disparities among various nation-states and populations exist . . . “
199-200: Not sure what resurfaces mean: “as he resurfaces memories of both violence and resilience.”
292: “the most popular. . . “ add “the”
387: “and complicit” not “complicity”
Addition of gender analysis in novel: nail salon labor as the consequence of war, now as refugees, V. women’s bodies are assimilated through this gendered immigrant laboring economy. How does this present as a re-narration that challenges the US exceptional war narrative, as it is no longer a dragging into the present? Or is it? This gendered labor is also a consequence of the silent, and repressed experiences of Vietnamese women refugees, who, as a result of broken and missing family (many Vietnamese men lost to war), must sustain their families and communities through this consigned low wage and labor intensive work. It might be useful to remind the readers of this, or how this and his experiences of queerness demonstrate "refugeetude" to keep the thread the theory continuous.
Comments on the Quality of English Language
118-19: the sentence doesn’t quite make sense, “much like economic disparities among various nation-states and populations exist . . . “
199-200: Not sure what resurfaces mean: “as he resurfaces memories of both violence and resilience.”
292: “the most popular. . . “ add “the”
387: “and complicit” not “complicity”
452-53: “In this way, the uneven 452 sexual dynamics operating between them point both at present discourses of sexuality 453 and prior and ongoing histories of racism.” “. . . point to both the present . . ,”
Author Response
RESPONSE TO REVIEWER 3:
- Summary
Thank you very much for taking the time to review this manuscript. Please find the detailed responses below and the corresponding revisions in track changes in the re-submitted file.
- Response to the reviewer’s comment:
Comment:
I just have a few edits and comments:
118-19: the sentence doesn’t quite make sense, “much like economic disparities among various nation-states and populations exist . . . “
199-200: Not sure what resurfaces mean: “as he resurfaces memories of both violence and resilience.”
292: “the most popular. . . “ add “the”
387: “and complicit” not “complicity”
452-53: “In this way, the uneven 452 sexual dynamics operating between them point both at present discourses of sexuality 453 and prior and ongoing histories of racism.” “. . . point to both the present . . ,”
Addition of gender analysis in novel: nail salon labor as the consequence of war, now as refugees, V. women’s bodies are assimilated through this gendered immigrant laboring economy. How does this present as a re-narration that challenges the US exceptional war narrative, as it is no longer a dragging into the present? Or is it? This gendered labor is also a consequence of the silent, and repressed experiences of Vietnamese women refugees, who, as a result of broken and missing family (many Vietnamese men lost to war), must sustain their families and communities through this consigned low wage and labor intensive work. It might be useful to remind the readers of this, or how this and his experiences of queerness demonstrate "refugeetude" to keep the thread the theory continuous.
Response to the reviewer’s comment:
All the reviewer's recommendations regarding the use of the English language have been integrated into the revised manuscript.
As per the addition of gender analysis in the novel, several revisions have been made to the final paragraph of section "3.1. No Refuge: Trauma and Neoliberal Exploitation" to incorporate the ideas highlighted by the reviewer. While the limitations of word count prevented a more exhaustive analysis, endeavors have been undertaken to capture the essence of the topic.
